# Complex hazard cascade culminating in the Anak Krakatau sector collapse

Thomas R. Walter [1], Mahmud Haghshenas Haghighi [1,2], Felix M. Schneider [1], Diego Coppola [3], Mahdi Motagh [1,2], Joachim Saul[1], Andrey Babeyko[1], Torsten Dahm[1], Valentin R. Troll [4,5], Frederik Tilmann[1,6], Sebastian Heimann[1], Sébastien Valade[1,7], Rahmat Triyono[8], Rokhis Khomarudin[9], Nugraha Kartadinata[10], Marco Laiolo [3], Francesco Massimetti[3,11] & Peter Gaebler[12]

Flank instability and sector collapses, which pose major threats, are common on volcanic islands. On 22 Dec 2018, a sector collapse event occurred at Anak Krakatau volcano in the Sunda Strait, triggering a deadly tsunami. Here we use multiparametric ground-based and space-borne data to show that prior to its collapse, the volcano exhibited an elevated state of activity, including precursory thermal anomalies, an increase in the island's surface area, and a gradual seaward motion of its southwestern flank on a dipping décollement. Two minutes after a small earthquake, seismic signals characterize the collapse of the volcano's flank at 13:55 UTC. This sector collapse decapitated the cone-shaped edifice and triggered a tsunami that caused 430 fatalities. We discuss the nature of the precursor processes underpinning the collapse that culminated in a complex hazard cascade with important implications for the early detection of potential flank instability at other volcanoes.

[1] GFZ German Research Centre for Geosciences, Helmholtz Centre Potsdam, Telegrafenberg, 14473 Potsdam, Germany. [2] Institute of Photogrammetry and GeoInformation, Leibniz University Hannover, 30167 Hannover, Germany. [3] Dipartimento di Scienze della Terra, Università di Torino, Via Valperga Caluso 35, Torino 10125, Italy. [4] Department of Earth Sciences, Uppsala University, Villavägen 16, Uppsala 75236, Sweden. [5] Faculty of Geological Engineering, Universitas Padjajaran (UNPAD), Jatinangor 45363 Bandung, Indonesia. [6] Institute for Geological Sciences, Freie Universität Berlin, Malteserstraße 74-100, Berlin 12249, Germany. [7] Department of Computer Vision & Remote Sensing, Technische Universität Berlin, Marchstr. 23, Berlin 10587, Germany. [8] Earthquake and Tsunami Center, Indonesian Agency for Meteorology, Climatology and Geophysics (BMKG), Jl. Angkasa 1 No. 2, 10610 DKI Jakarta, Indonesia. [9] LAPAN, Remote Sensing Application Center, Jl. Kalisari N0. 8, Pekayon Pasar Rebo, Jakarta, Indonesia. [10] Volcano Research and Monitoring Division, CVGHM - Geological Agency of Indonesia, Jl. Diponegoro No. 57, Bandung 40228, Indonesia. [11] Dipartimento di Scienze della Terra, Università di Firenze, Via La Pira 4, Firenze 50121, Italy. [12] BGR, Federal Institute for Geosciences and Natural Resources, Stilleweg 2, Hannover 30655, Germany. Correspondence and requests for materials should be addressed to T.R.W. (email: twalter@gfz-potsdam.de)

Volcanic islands are typically fast-growing edifices that rest on a complex morphology and weak substrata[1], and they are frequently made up of highly fragmented, mechanically unstable material[2]. Therefore, many volcanic islands rise rapidly but also erode swiftly via volcano flank instability[3], leading to irregular shapes and embayments owing to sector collapses[4]. Geomorphic amphitheaters are common subaerial remnants of fast lateral landslides; these events often recur at the same location[1]. They may result in distal run-out submarine deposits, which demonstrate the intense dynamics of tsunami-genic mass movements in the oceans[5]. In fact, data collected from around the world reveal that historical volcano-induced tsunamis have caused significant damage and loss; ~130 events have been recorded from 80 different source volcanoes since 1600 AD[6–8]. These events have been caused by the entry of pyroclastic flows into the ocean[9–11] and their submarine continuation[12], by caldera collapse[13], and by landslides entering the ocean[5,14,15], or combinations thereof[6,7,16]. Among the progenitors of these events, 17 historically identified source volcanoes are located in Southeast Asia[17]. Volcano-induced tsunamis have probably led to the demise of ancient civilizations[18,19] and are responsible (albeit uncommonly, i.e., 5% of all tsunami events) for approximately one-fourth of all fatalities attributed to volcanic activity[3,6,20].

It has been over 135 years since the infamous volcano-induced Krakatau tsunami occurred on 27 August 1883 (Fig. 1a). A common problem of such events is that they are rare[16], and thus, although volcanic islands introduce numerous recognizable threats such as instability, sector collapse and tsunamis[21,22], little is known about their precursor activity and possible strategies to mitigate the associated risks. Moreover, the preparation and initiation of sector collapses are complex, insomuch that they could possibly be associated with faulting, slumping, and pyroclastic flows or combinations thereof[23–26]. Consequently, at present, no consensus exists regarding what constitutes a reliable precursor signal for sector collapse on a volcanic island[10,20].

During the 1883 Krakatau eruption and tsunami, which is estimated to have killed over 36,000 people, 12 km[3] of dense rock equivalent was erupted; a caldera collapse occurred as a result, leaving only small and steep subaerial remnants of the former volcano edifice along the rim of a 7-km-wide deep-water caldera basin[27,28]. Volcanism continued after the 1883 events, eventually producing Anak Krakatau ("the son of Krakatau"), where several additional smaller tsunamis were triggered at this site by processes such as underwater explosions[29]. It is possible that the island known as Anak Krakatau is preconditioned for landslide-triggered tsunamis, as it is situated on a steep morphological cliff. This edifice first breached the sea surface in ~1928 and gradually formed a 150-m-high tuff ring by 1959[22]. A gradual shift in activity then occurred toward the southwest, resulting in further growth of the edifice over the cliff and toward the deep submarine caldera basin[27]. This was leading to recent concerns about a possible landslide from the southwestern island flank and the corresponding generation of a tsunami[22]. These concerns and scientific assessments turned into reality following an intense (but

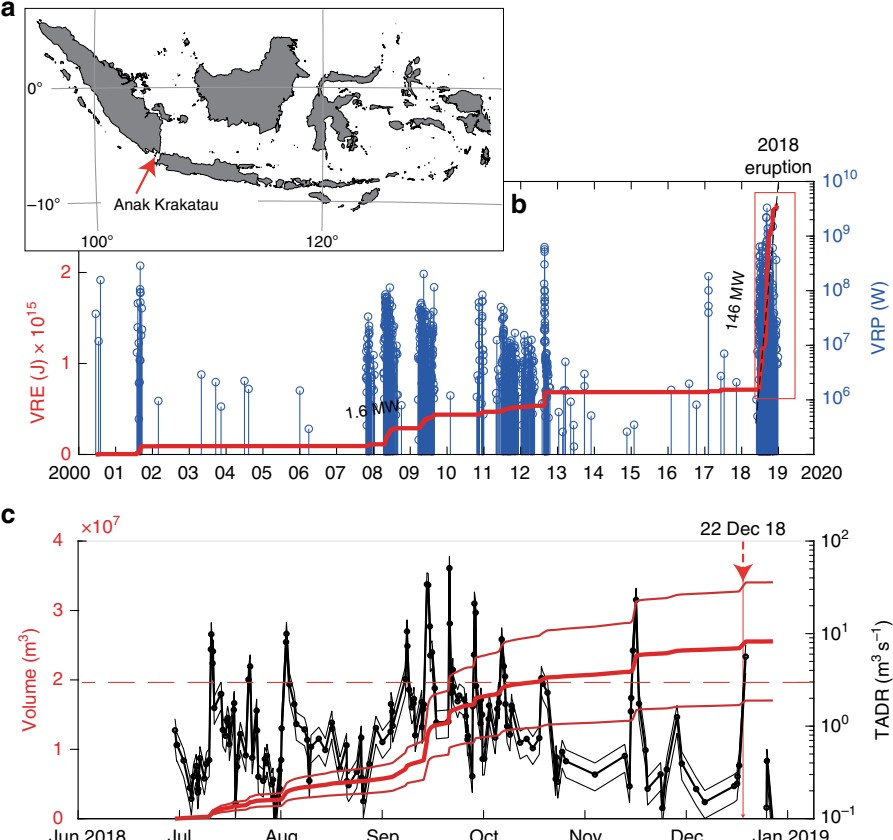

**Fig. 1** Elevated volcanic activity prior to sector collapse. **a** Location of Anak Krakatau. Coastline map created using the GMT/MATLAB Toolbox[59]. **b** Multiyear time series of the volcanic radiative power (VRP) and cumulative volcanic radiative energy (VRE) recorded as MODIS data at Anak Krakatau. A sharp increase in thermal activity started on 30 June 2018, marking the beginning of a new eruptive phase that culminated in the collapse of the edifice on 22 December 2018. The red rectangle marks the recent eruptive period shown in detail in Fig. 3. **c** Time-averaged discharge rate (TADR) derived from satellite thermal data during the June–Dec 2018 eruption. Note the occurrence of 11 pulses with a TADR above $3\,\mathrm{m^3\,s^{-1}}$ (dashed line) associated with effusive paroxysms that produced lava flows. The cumulative erupted volume (red line) indicates a gradual decline in effusive activity after Oct 2018

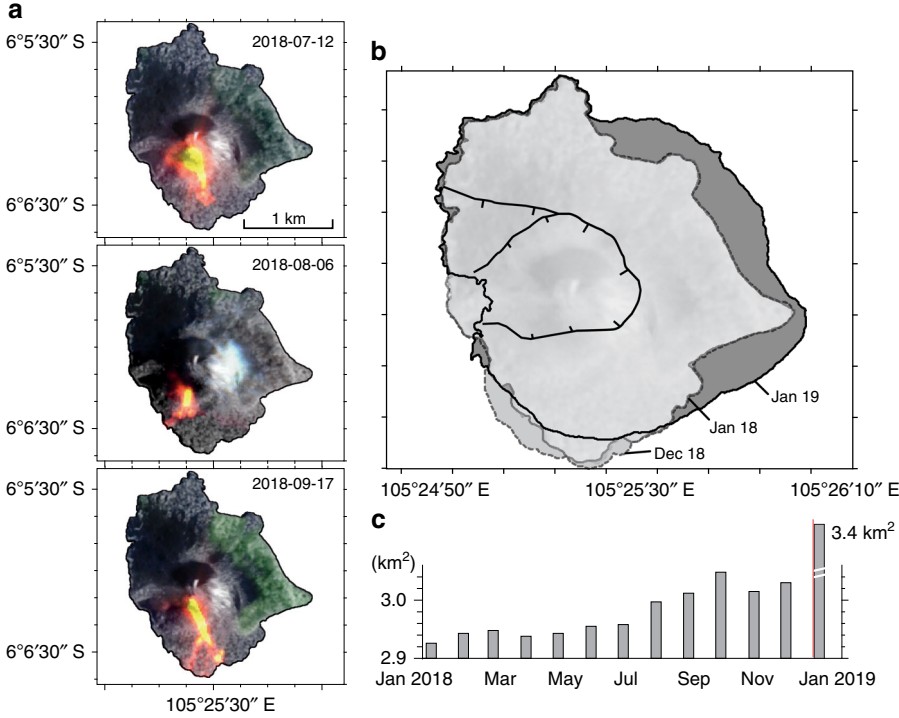

**Fig. 2** Eruptions and island perimeter growth map. **a** Several selected Sentinel-2 images (band combination of 12, 11, 4) showing the emplacement of hot and new material (red-yellow) on the southern flank of Anak Krakatau during the increased eruptive activity in 2018 (see also Supplementary Figure 1). **b** Island perimeter maps derived from satellite radar amplitude images (Supplementary Figure 2) show little variation from January 2018 to June 2018, followed by southward growth from June 2018 to December 2018 prior to the sector collapse (light gray). The black lines indicate the new scarps formed by the sector collapse. The outer outline indicates the postcollapse island perimeter in January 2019 (dark gray). **c** Land area change based on monthly island perimeter analysis. The area changed gradually prior to the flank collapse, but more rapidly after the collapse event. Maps created using the GMT/MATLAB Toolbox[59]

theretofore unidentified) increase in precursor activity. Although flank motion was identified[30,31], the hazard was not systematically monitored. On 22 December 2018, this volcanic center once again became the source of a tsunami that struck the highly vulnerable Sumatran and Java coasts. According to the Indonesian National Disaster Management Authority (BNPB), the 22 December 2018 tsunami caused over 430 fatalities, injured 14,000 people, and displaced 33,000 more along the Sunda Strait. The tsunami risk of this area is particularly high as the coast is very popular with both locals and tourists and is home to >20 million people within a 100-km distance from the volcano[22].

By combining different ground and satellite data we can outline the details of the complex hazard cascade leading up to the events on the 22nd December 2018. Our study reveals that Anak Krakatau showed clear signs of flank motion and elevated volcanic activity prior to sector collapse which triggered the destructive tsunami.

## Results

**Preparation of the flank collapse event**. Satellite monitoring and ground observations reveal that Anak Krakatau was in an elevated stage of activity throughout 2018. An analysis of infrared data recorded by the thermal sensors of the Moderate Resolution Imaging Spectroradiometer (MODIS)[32] indicates that a new intense eruptive phase initiated at Anak Krakatau on 30 June 2018 (Fig. 1b). This eruptive phase was the most intense recorded since systematic data acquisition began in 2000 and was characterized by a mean volcanic radiative power (VRP) of ~146 MW, which is ~100 times the long-term thermal emission (~1.6 MW) recorded between 2000 and June 2018 (Fig. 1b). This thermal

activity was associated with persistent Strombolian to Vulcanian activity and the emplacement of eruptive deposits along the center and the western and southern flanks of the volcano (Supplementary Figure 1). This eruptive phase continued for 175 days until 22 December 2018, when the activity suddenly evolved into a sector collapse.

An estimate of the erupted volume derived from thermal data (see methods) indicates that the eruption phase produced $25.5 \pm 8.4 \, \text{Mm}^3$ of deposits, implying a mean output rate of $1.7 \pm 0.8 \, \text{m}^3 \, \text{s}^{-1}$ from June 2018 to just prior to the collapse event. Thus, the load acting on the summit and especially the southern flanks of the island progressively increased over this time by ~54 million tons (assuming a mean density of $2110 \, \text{kg} \, \text{m}^{-3}$)[33]. The 2018 eruptive period was punctuated by 11 pulses with time-averaged discharge rates (TADRs) higher than $3 \, \text{m}^3 \, \text{s}^{-1}$. The occurrence of these effusive pulses peaked between September and October 2018, with the three highest TADRs of 10.5 (±3.5), 33.4 (±11.1), and 50.9 (±16.8) $\text{m}^3 \, \text{s}^{-1}$ on 9, 15, and 22 September 2018, respectively (Fig. 1c). Starting in October 2018, the rate of these pulses declined, both in intensity and in frequency, except for a period in mid-November with a peak TADR of 23.2 (±7.6) $\text{m}^3 \, \text{s}^{-1}$. The general decrease in activity after mid-October is also suggested by the trend in the development of the cumulative volume of erupted materials (red line in Fig. 1c).

According to satellite images from the European Sentinel-2 mission (Fig. 2a), at least $0.85 \, \text{km}^2$ of the island (28% of the total area) was covered with abundant hot ejecta and new deposits (Supplementary Figure 2). Many of these entered the sea, adding $0.1 \, \text{km}^2$ of land surface to the southern shore of the island (the island area increased from 2.93 to $3.03 \, \text{km}^2$) by early December 2018, as assessed by shoreline edge detection analysis (Fig. 2b,

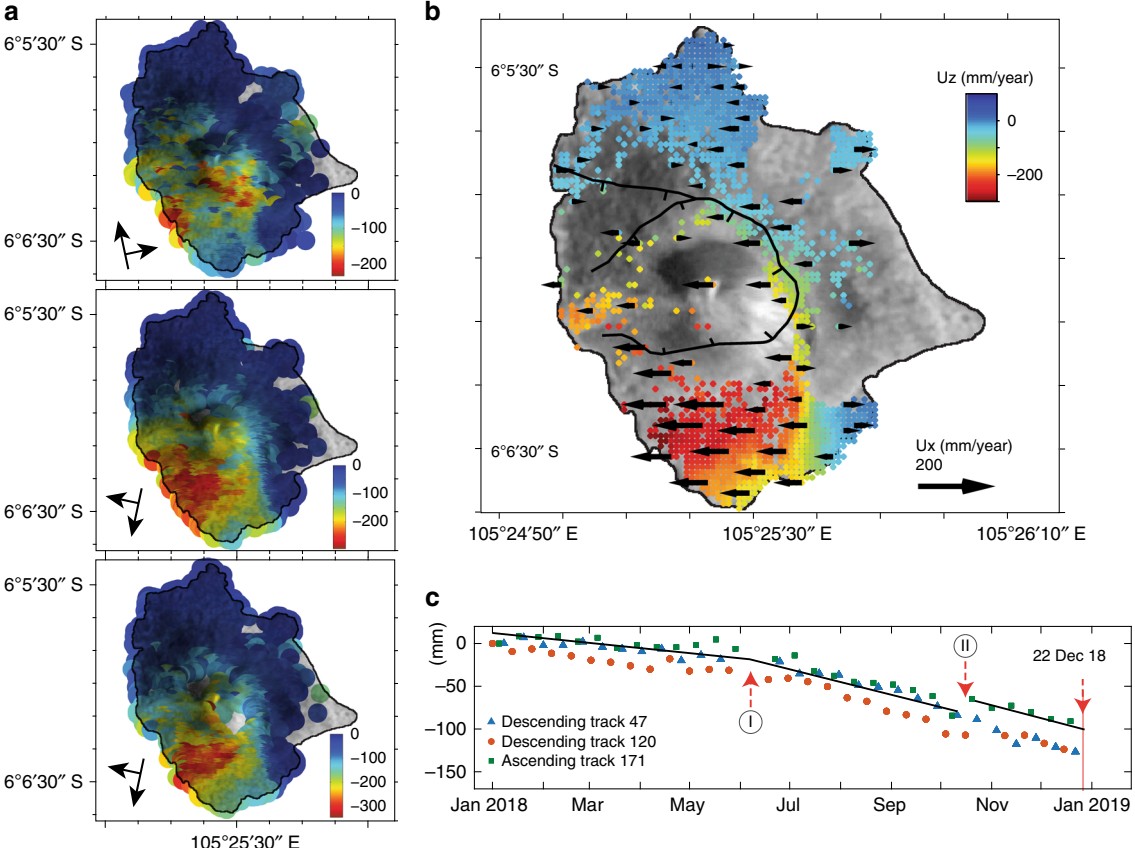

**Fig. 3** Deformation maps and island perimeter changes. **a** InSAR time series showing the movement of the ground in the satellites' line-of-sight (LOS) direction, generated for the period between 1 January and 22 December 2018 for one ascending and two descending tracks. The different viewing geometries reveal the deformation of the southwestern flank of Anak Krakatau. **b** Average vertical (colors, red is down) and E-W (black vector symbols) movement computed from the different InSAR viewing geometries. **c** Deformation trends for the different tracks (average pixel values in the inner sector collapse scarp outline) showing pronounced trend changes in June–September 2018, labeled I and II, and identifying the 22 December 2018 collapse event. Maps created using the GMT/MATLAB Toolbox[59]

Supplementary Table 2). A compositional analysis of ash sampled during the intense eruption phase on 22 July 2018 indicates a basaltic andesite composition (Supplementary Table 1), which overlaps with the typical compositional spectrum displayed by Anak Krakatau in recent decades (see Supplementary Figure 3).

Interferometric synthetic aperture radar (InSAR, Supplementary Figure 4) analysis and time-series analysis (Fig. 3a, Supplementary Figure 4) show that the southwestern and southern flanks of Krakatau were already slowly subsiding and moving westward at the beginning of our analysis window in January 2018 (Fig. 3b), despite the absence of significant thermal anomalies. The deformation that occurred in the subsequently collapsed sector was advancing at an approximately constant rate with a peak of almost 20 mm (or ~4 mm per month) in the satellites' line-of-sight direction until the volcanic activity increased in late June 2018, at which point the deformation markedly accelerated (labeled "I" in Fig. 3c; ~10 mm per month). In addition, short-term eruptive pulses in Sep–Oct 2018 resulted in a minor step change (labeled "II" in Fig. 3c). Therefore, the data show that increased eruption rates coincide with increases in flank movement. An analysis of the deformation field pattern reveals that it affected over one-third of the island, exhibiting a moderate gradient on the west side and a well-identified gradient on the southeast side (Supplementary Figure 5). The cumulative deformation pattern indicates a progressively sliding flank that can be explained by a deep décollement plane, simulated as a

rectangular dislocation[34], with a dip of 35°, a strike of 163°, and a slip of 3.36 m (Supplementary Figures 6–7). Notably, deformation also affected the island summit, and therefore potentially shearing its main magmatic and hydrothermal-plumbing systems.

The dynamics of the moving flank were relatively slow; as a consequence, seismic stations installed on the mainland for tsunami early warning were hardly able to record this type of movement. Then, conditions started to change shortly before the sector collapse event. Satellite thermal data show a pulse ($5.6 \pm 1.9 \ m^3 \ s^{-1}$) on 22 December 2018 at 06:50 UTC, just a few hours before the onset of the collapse (Fig. 1c). Compared with the thermal pulses recorded earlier in 2018, this eruption was relatively small. Infrasound records show the release of continuous high-frequency energy (0.5–5 Hz) from Krakatau, indicating high levels of volcanic activity in the hours prior to the collapse followed by a brief period of quiescence (Supplementary Figure 8). Both the intense activity earlier in the day and the quiet period were further confirmed by eyewitness accounts. Seismic stations (Fig. 4a) then suddenly recorded a high-frequency event (2–8 Hz, Fig. 4b), just ~115 s before the flank collapsed on 22 December 2018 (marked "1" in Fig. 4c), representing the last and most immediate precursor—or even trigger—of the main sector collapse (marked "2" in Fig. 4c). The seismic signal originated at Anak Krakatau and was associated with either an earthquake (local magnitude $M_L = 2$–3) or an explosion with seismic amplitudes that exceeded even those of the sector collapse in the 4-8 Hz frequency band (Fig. 4c) and was even recorded by

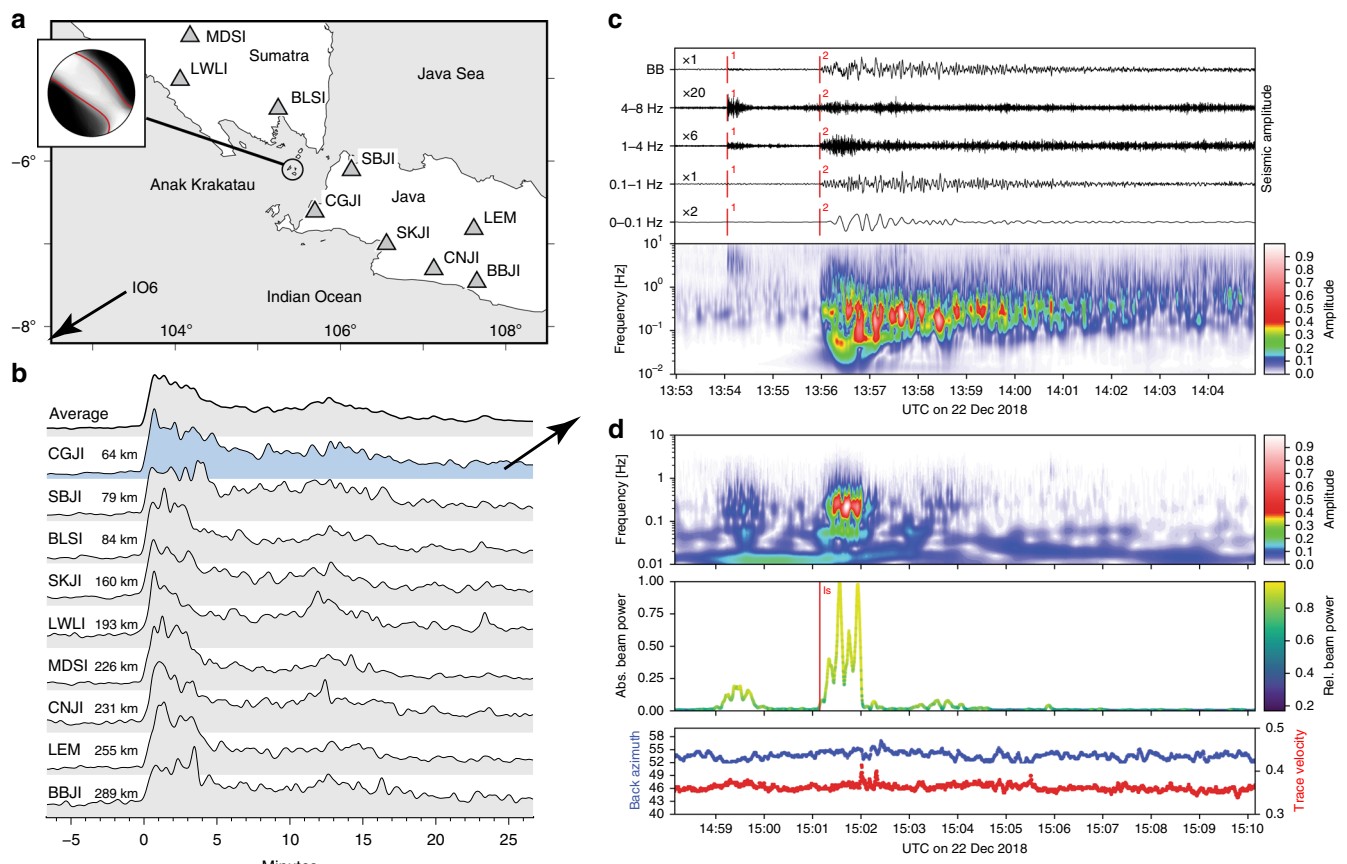

**Fig. 4** Seismic and infrasound recordings of the Anak Krakatau sector collapse. **a** Locations of stations with regional seismic instruments. Station I06AU is an infrasound station located 1150 km to the SW of Krakatau (beyond the edge of the map). The inset shows the probabilistic moment tensor solution with the best nodal plane striking NW–SE and dipping to the SW and a large compensated linear vector dipole (CLVD) component. The red line indicates the nodal surface of the best double couple component from the full moment tensor solution. **b** Smoothed envelopes of vertical component 0.4–1 Hz bandpass-filtered seismic records. **c** Normalized seismic amplitudes at the closest station (CGJI), located 64 km from Anak Krakatau, revealing the occurrence of a high-frequency event (1) 115 s prior to the sector collapse (2). The spectrogram reveals that collapse is a 1–2-minute-long low-frequency signal presumably related to the landslide, followed by ~5 mins of strong emissions at high frequencies. **d** Upper and middle panels: Infrasonic spectrograms of the best beam and beam power from the seven-element infrasound array at I06AU, where the onset of a strong impulsive signal was registered at 15:01:09. The beam direction and trace velocity, displayed in the lowermost panel, point towards Krakatau (back azimuth 55–56°, compared with Supplementary Figure 8) with a trace velocity of ~60 m s⁻¹. Although the strong impulsive signal (the landslide) spanned a duration of only ~1 min, in the time window shown in the figure, the beam focuses continuously towards Krakatau, indicating continuous energy flow from Krakatau for a much longer time span (see also Supplementary Figure 8). The red mark (labeled *ls*) shows the theoretical arrival time of the stratospheric phase predicted by infrasound modeling, taking into account the origin time of the seismic event at 13:55:49 at Krakatau

infrasound stations at large distances (Fig. 4d). The coda (1–8 Hz) of this event is unusually long compared with those of tectonic earthquakes of comparable magnitude; in fact, the coda is still discernible when the onset signal of the catastrophic sector collapse becomes identifiable (Supplementary Figure 9).

**The catastrophic event**. Local, regional, and even some teleseismic seismic stations (Fig. 4a) show clear signatures of the tsunami-triggering event. The abrupt onset of a short-period seismic signal is followed by ~5 mins of strong emissions at 0.1–4 Hz, approximately coinciding with a long-period signal (0.01–0.03 Hz) occurring over a shorter duration (~90 s) that we interpret as the seismic signature of the main mass movement of the landslide (Fig. 4c). The onset times of the short-period signals at stations in Sumatra and Java are consistent with the location of the volcano and an origin time of 13:55:49 UTC (Fig. 4d). The inversion of low-pass filtered (0.01–0.03 Hz) surface waves reveals an event with a moment magnitude of 5.3 (Supplementary

Figure 10). A significant non-double-couple component is retrieved from the inversion of low-pass filtered seismograms, indicating a linear vector dipole oriented to the SW at 222° and a dip angle of 12° (or alternatively representing tensile opening mixed with a shear rupture dipping ~61° to the SW) (Supplementary Figure 11). As these parameters are close to those of the pre-eruptive décollement plane derived from InSAR data (NW–SE strike and SW dip), we conjecture that it was this plane that constituted the failure plane during the sector collapse. Following the main event, a nearly continuous tremor-like signal exhibiting a slowly decreasing intensity with frequencies 0.7–4 Hz (Fig. 4c) was recorded at all nearby stations and was attributed to strong volcanic explosions.

The effects of this event were recorded extensively. The Australian infrasound array (I06AU) located over 1150 km to the SW of Anak Krakatau recorded a high-energy impulse at 15:01:09 UTC, which translates to a modeled origin time of 13:55:49 (±4 s) UTC at the Krakatau site (Fig. 4d, Supplementary Figure 12). This timing is consistent with the origin time of the short-period

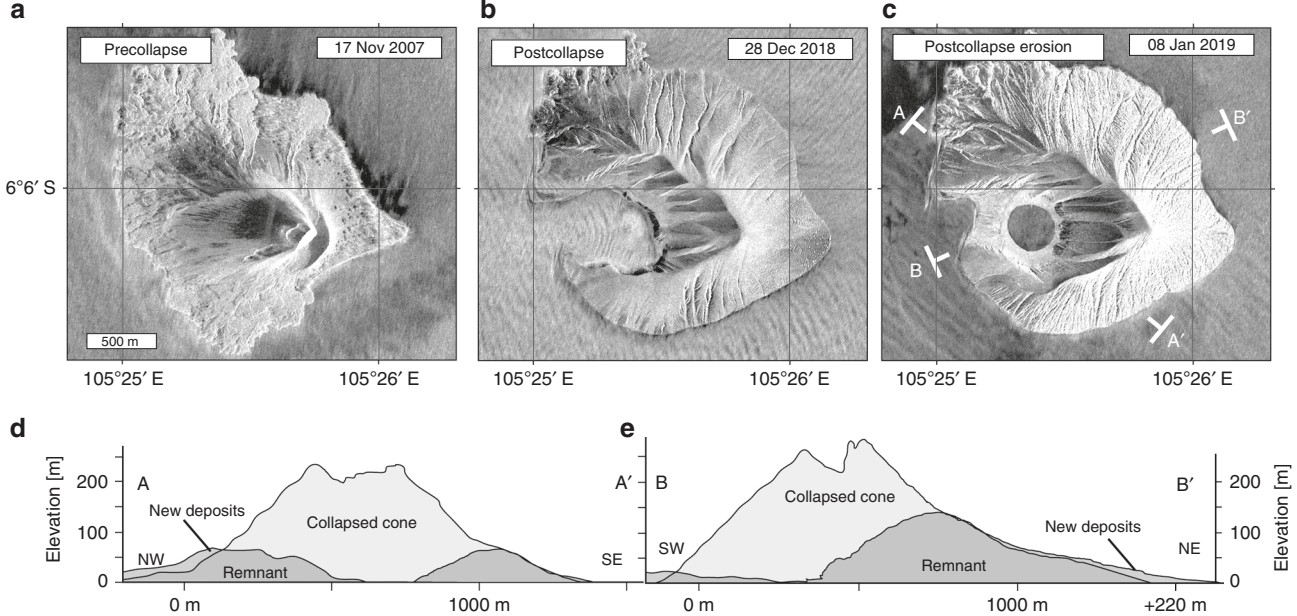

**Fig. 5** Morphological changes by sector collapse, material deposition, and subsequent erosion. **a–c** TerraSAR-X satellite radar images in high-resolution spotlight mode showing the extents of changes associated with sector collapse and with the formation and erosion of the new coastline, tuff ring, and crater lake before and after the 22 December 2018 collapse. **d, e** Before and after comparison of the morphology deduced from TanDEM-X data (before, light-gray shading) and camera drone data (after, dark gray shading) revealing profound topographic changes, material loss and deposition of new materials. Erosion was carving gullies on the volcano flanks by early January 2018

seismic signal at Anak Krakatau (identified as the landslide signal). The duration of the impulse is broadly comparable to the long-period seismic signal and indicates that subaerial sliding lasted for ~1-min only (Fig. 4d). Both the seismic records at local stations and the infrasound records from the I06AU array (Supplementary Figure 12) continued to be dominated by coherent emissions from Anak Krakatau (presumably related to strong volcanic eruptive activity there) for at least several hours. The dominant frequency of the eruption signature in the infrasound signal shifted by nearly an order of magnitude (from ~0.8–4 Hz prior to the landslide to 0.1–0.7 Hz afterward). Even the closest local stations did not pick up a clear signature of any prelandslide eruption, but postlandslide eruptions dominated the seismograms of stations even a few hundred kilometers away. Together, these observations suggest a profound change in eruptive style following the landslide. Furthermore, on 23 December 2018 at 06:31 UTC, a large $SO_2$ cloud was detected (Supplementary Figure 13), likely resulting from the decapitated and degassing hydrothermal system. In contrast, no similarly strong degassing was detected in the weeks prior to the flank collapse event.

Tsunami arrivals were recorded by four tide gauge stations on the Sumatra and Java coasts (Supplementary Figure 14). Back-tracing from these four stations, using the classic tsunami travel time approach (see methods), suggests that the source location corresponds to the southwestern part of Anak Krakatau and that the source origin time corresponds to that revealed by the broadband seismic analysis. Therefore, the backtracing simulation shows that the tsunami was triggered by the long-period landslide during the first minutes of the event and not by the following volcanic eruptions.

The full extent of the sector collapse event initially remained hidden owing to intense postcollapse eruptive activity but became visible when the eruption intensity decreased again by 27 December 2018. As a result, a new and steep amphitheater enclosing a deep valley became distinguishable on the

southwestern sector of the island. The deposition of new material shifted the coastlines. The collapsed area is readily identified in satellite radar imagery (Fig. 5a) and is located in the area that was subsiding and moving laterally outward prior to the collapse event (Fig. 3). The area affected by landslides, however, is smaller than the area affected by precursory deformation; accordingly, we estimate that only 45–60% of the deforming subaerial flank actually failed. High-resolution camera drone records in January 2019 allow the partial derivation of a digital elevation model (Supplementary Figure 15). By comparing the digital elevation models from before and after the event, we ascertain that the sector collapse reduced the height of the island from 320 to 120 m (Fig. 5) and removed the former edifice peak, thereby decapitating the main eruption conduit (Fig. 6). Detailed volumetric estimates obtained upon differencing the two digital elevation models suggest an estimated volume loss of $1.02 \times 10^8$ m³, which is a minimum estimate, as it does not consider the volume gained by new eruptive deposits (which may exceed another $1 \times 10^8$ m³, Supplementary Figure 16); furthermore, the submarine collapse volume is not included and necessitates forthcoming bathymetric surveys. Tephra deposition occurred immediately after the sector collapse (between 22 and 25 December 2018, as determined by satellite radar images), causing a shift in the perimeter of the island and overprinting the collapse scar geometry.

Profound changes continued to occur in the weeks following the catastrophic event. Numerous small slumps deposited material into the landslide amphitheater and an explosion tuff ring formed inside the decapitated volcano conduit area. The eruption site now appears slightly shifted to the SW, hosting a new 400-m-wide water-filled crater (Fig. 5c). Thermal activity was detected after the collapse, possibly linked to ongoing eruptions. Although the collapse of the southwestern sector into the ocean was associated with a considerable volume loss, area calculations of the island reveal rapid regrowth (over 10%) from December 2018 to January 2019 (Fig. 2b), which was mainly associated with the (re-)deposition of pyroclastic material.

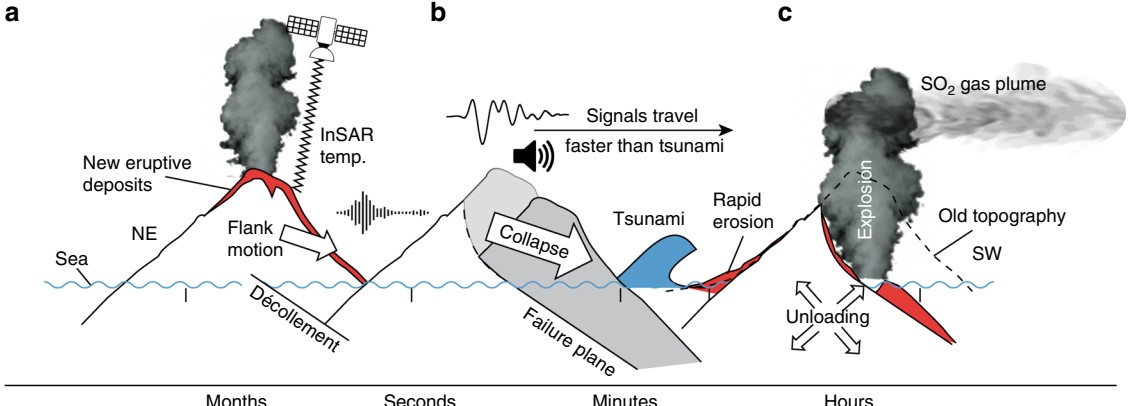

**Fig. 6** Cascade of precursors leading up to the 22 December 2018 sector collapse event. **a** Precursors include flank motion (white arrow), eruptions (represented as eruption cloud), and increasing eruptive deposits (red shaded areas) as assessed by satellite data (thermal, InSAR). A décollement (black line beneath the island) dips SW, but faulting had not yet breached the surface. Approximately 2 mins before the collapse, a seismic event was recorded (shown by seismic trace symbol). **b** The landslide collapse along a failure plane (black curved line beneath island) showed a 1–2-min-long low-frequency signal (seismic waveform). Infrasound instruments (speaker symbol) measured the collapse before the tsunami waves arrived. The collapse decapitated the island (gray shaded area). The tsunami (blue wave) caused damage and loss along the coast. **c** Postcollapse volcanic explosions occurred coincident with increased degassing (gray plume) caused by unloading (arrow symbol); the old topography is indicated (black dashed line). New eruptive deposits increased the island area (red shaded areas). Finally, rapid erosion deeply carved incisions into the fresh eruption deposits

## Discussion

On the basis of the remotely sensed displacement data and thermal analysis, we conclude that the Krakatau volcano showed clear signs of flank motion and elevated volcanic activity prior to the 22 December 2018 sector collapse. The long-term hazard owing to Anak Krakatau's steep volcanic edifice had already been described, and thus, the collapse event and subsequent tsunami were anticipated hazards[22]. The month-scale precursors included the strongest thermal activity recorded at Anak Krakatau in ~20 years and an accelerated flank motion; these characteristics made Anak Krakatau one of the most rapidly deforming volcano flanks known on Earth prior to its collapse (Fig. 3). In fact, deformation was already identified along the southwestern flank of Anak Krakatau in InSAR time series over 10 years before December 2018[30,31], but this deformation was not interpreted to be a potential precursor of a larger sector collapse. Compared with other volcanoes that exhibited flank deformation prior to sector collapse, the movement at Anak Krakatau also corresponded with eruption pulses, possibly associated with pressure changes in the volcano interior, and therefore Anak Krakatau shares a similar behavior with volcanoes elsewhere[35,36].

We investigated whether changes in composition could explain the increase in magma production at Anak Krakatau prior to its collapse, but our analysis of syn-deformation tephra samples suggest that the material was not significantly different from the material erupted in past decades, implying that deep magmatic changes were likely not directly responsible for the observed dynamic changes at the surface. The orientation of the main sliding plane of the collapse event could be identified from the seismic records of the collapse, suggesting a steeply southwesterly dipping failure nodal plane. The strike and dip of this plane are geometrically in agreement with the amphitheater morphology and also notably with the inferred dislocation plane of precursory creep motion. Therefore, we conclude that the landslide décollement had already developed before the collapse.

Furthermore, as a significant proportion of the island and its shallow plumbing system was removed by the flank collapse, unloading may affect (also future) postcollapse compositions[35,37,38], the structural evolution, magma pathways, and eruption locations[39,40].

A remaining question is whether the landslide of Anak Krakatau was triggered by volcanic or seismic activity. Our observations indicate that the climax of the eruptive phase was recorded in late September 2018, ~3 months prior to the flank collapse. Indeed, from September to December 2018, the volume of newly deposits followed a generally decreasing trend (Fig. 1c). In addition, $SO_2$ gas emissions were low in the weeks prior to the collapse (Supplementary Figure 13). Moreover, because the deformation rate remained almost constant throughout this period (Fig. 3c), we suggest that only minor change, if any, was attributable to the accumulation of magma into the shallow portions of the edifice. However, the intense activity witnessed throughout the year likely increased the overall instability of the edifice owing to the rapid accumulation of new material. In fact, studies elsewhere show that slope instability at volcanoes is not always associated with eruptive phases[6]. This relationship is observed because slope instability changes over time; in addition, fault planes and other zones of weakness are strongly affected by pore pressure changes, hydrothermal activity, and mechanical weakening by alteration, as well as by sea erosion and oversteepening[2,3,6,41]. A similar but much smaller sequence recently occurred also at Mount Etna, where a short-term increase in the magma supply and eruption rate was accompanied by magmatic intrusion, leading to the collapse of an unstable cone[42]. This example demonstrates that under such critical conditions, minor internal and external perturbations can potentially trigger a collapse and eruption, leading to a disaster. From this perspective, our hypothesis that the seismic event identified herein 2 mins before the Anak Krakatau landslide acted as an external trigger is plausible but remains to be tested further.

Volcano-induced tsunamis are thought to be rare and are therefore not commonly considered in tsunami early warning centers. Historic documents reveal, however, that Southeast Asia experiences volcano-induced tsunami hazards relatively frequently, with 17 events during the 20th century and at least 14 events during the 19th century[7], defining a recurrence rate of one event every 5–8 years. A volcano-induced tsunami from Anak Krakatau was anticipated[22], but accurate predictions were impossible owing to a lack of understanding of the processes involved. Hence, the study of the 22 December 2018 sector

collapse at Anak Krakatau provides us now with important information about the precursors and processes that culminated in the disaster.

The tsunami reached the coastal towns of Jambu, Ciwandan, Agung, and Panjang within 31, 38, 39, and 57 mins, respectively. The tsunami waves were overtaken by the faster seismic waves and the infrasound signals of the strong explosive eruption (Fig. 6b), that were associated with the landslide and decapitation of the hot interior of the volcano followed by steam-driven phreatomagmatic explosions (Fig. 6c).

In fact, the seismic records following the sector collapse event indicate that tremor activity continued for hours, resembling the volcanic tremors associated with steam-driven explosions elsewhere[43], although the large distance between the volcano and the seismic network may have blurred this interpretation. The eruptive style of the postdecapitation eruptions was different from that of the predecapitation eruptions. This is indicated by the different frequency contents in the seismic and infrasound data and the onset of significant $SO_2$ emissions on 22–23 December 2018, in agreement with the period characterized by a reduction in radar amplitudes, the deposition of erupted material and a shift in the coastline, as seen in the ground-range detected (GRD) images on 22–25 December 2018. The productivity of the eruptions also appeared to increase, as evidenced by the marked growth of the island area after the collapse. This suggests a major effect of unloading on the magmatic and hydrothermal interior of the volcano.

It appears that a perfect storm of magma-tectonic processes at Anak Krakatau culminated in the 22 December 2018 tsunami disaster. Leading up to the event, different sensors, and methods measured distinct anomalous behaviors, which in hindsight can be deemed precursory. However, at the time and when considered individually, none of the parameters, including the thermal anomalies, flank motion, anomalous degassing, seismicity, and infrasound data, were sufficiently conclusive to shed light on the events that were about to unfold.

This study demonstrates that volcano sector collapses and the resulting tsunamis might be effectively anticipated by continuously monitoring the various preparation stages. Long-term flank motion, changes in thermal emission, and short-term seismic events precede the collapse, which itself was well monitored by low-frequency seismic waveforms and infrasound stations. Assessments of these parameters could be implemented in available early warning systems. Therefore, the next-generation tsunami early warning system must consider multiparametric observations, since our study reveals that a multitude of changes signified an unprecedented level of activity at Anak Krakatau prior to 22 December 2018. We hence recommend a dedicated search for island volcanoes that are susceptible to flank collapses and those with a history of tsunamis, and we advise the development of appropriate monitoring programs to identify critical systems at these sites.

Because the Anak Krakatau sector collapse and tsunami are rare events, insights such as those reported herein yield vital information on precursor processes and aid in refining existing monitoring and early warning technologies.

## Methods

**Satellite thermal data**. Satellite thermal time series were generated using Middle Infrared Observations of Volcanic Activity (MIROVA), a volcanic hotspot detection system based on the analysis of MODIS data[32]. Two MODIS sensors carried on board two NASA satellites, Terra and Aqua (in orbit since March 2000 and May 2002, respectively), provide approximately four images per day (two daytime and two nighttime) of the entire Earth surface with a nominal spatial resolution of 1 km$^2$ per pixel in the infrared band. Through a series of spectral and spatial processing steps[32], the MIROVA algorithm detects, locates, and quantifies any hotspots within an area of 2500 km$^2$ (50 × 50 km) surrounding the target volcano

and provides near-real-time estimates of the VRP, which represents the radiant heat flux (in Watts) emitted by the detected volcanic activity (www.mirovaweb.it). We excluded images acquired under cloudy conditions, those with a poor geometry (i.e., a high satellite zenith angle) and those showing nonvolcanic hotspots such as fires or false positives, which are sometimes produced by the automatic detection algorithm. The visual inspection allowed us to retrieve robust VRP time series, which were then used to estimate the time-averaged discharge rate (TADR; in m$^3$ s$^{-1}$) and erupted volume according to the following procedure[44]: TADR = VRP/$c_{rad}$, where the radiant density $c_{rad}$ (J m$^{-3}$) is an empirical parameter that embeds the rheological, topographic, and insulation conditions characterizing the observed lava flow. For the Anak Krakatau eruptive deposits we used $c_{rad}$ = 0.5–1 × 10$^8$ J m$^{-3}$, a range of values consistent with lavas having a basaltic to andesitic composition[44]. The integration of TADR values over time provides an estimate of the erupted lava volume with an uncertainty of ± 33% (based on the range of plausible $c_{rad}$ values).

**InSAR displacement analysis**. We measured surface displacements prior to the 2018 sector collapse through the InSAR time-series analysis generated from Sentinel-1 (S1) radar images in one ascending and two descending orbits (see Supplementary Table 3) using the StaMPS-MTI method[45], which is a freely available and widely used multimaster time-series analysis method (Supplementary Figure 5). The S1 Line-of-Sight (LOS) ground motion maps for the period between 1 January 2018 and 22 December 2018 were combined to calculate the vertical and horizontal motions before the eruption[46]. To describe the overall flank motion time series, we estimated the average displacement values for all pixels located in the white polygon outlining the subsequent sector collapse region. In this way, the possible contributions of the loading and compaction of new eruptive deposits, which overprint the landslide signal, are eliminated. The exact locations are identified in coherence maps and Sentinel-2 images (Supplementary Figure 4a, b). This average time series displays short-term transients i, labeled (I) and (II) in Fig. 3c. This estimate can be regarded as a minimum owing to the large displacement gradient in the region after June, leading to the underestimation of the true phase during the unwrapping process[45]. Furthermore, the mean velocity was modeled using dislocation theory to locate the décollement of the landslide. Inversion modeling of the same data suggests a sliding plane that dips 34 degrees to the southwest (Supplementary Table 4).

**Island perimeter**. The changes in the coastline of Anak Krakatau were mapped using S1 GRD scenes in the Google Earth Engine cloud computing environment[47]. The backscatter of all ascending and descending GRD images was first stacked on a monthly basis; then, to reduce speckle noise and increase the signal-to-noise ratio, we applied a low-pass temporal filter followed by a low-pass spatial filter. The stacked images were subsequently segmented using an adaptive threshold to separate land from water bodies. The changes in the morphology of the island following the eruption were investigated using three high-resolution spotlight images from the TerraSAR-X satellite before and after the eruption. These images were used to verify the coastline information obtained from the GRD scenes.

**Seismic analysis**. Waveforms from the Indonesian broadband seismic network in Sumatra and Java were scanned for high-frequency (0.3–1 Hz) signals and abrupt changes in the root-mean-square amplitudes of the traces. The first arrivals of the abrupt onset of the high-frequency signal were picked at nine stations; assuming that the first arrival corresponds to the P-wave phase generated by the onset of slope instability, the located epicenter coincided with Anak Krakatau. The local magnitude ($M_L$) of this event was 3.1. Strong long-period (LP) surface waves below 0.03 Hz could be recognized at stations at epicentral distances of several thousand kilometers. These LP signals lack clear high-frequency body wave onsets, as would be expected for typical ruptures associated with tectonic earthquakes. LP waves produced by landslides have been modeled by single force or linear force couples, where the uphill and downhill forces point in opposite directions and represent the acceleration and deceleration, respectively, of the sliding mass[48,49]. The linear vector dipole model is included for a full moment tensor description. We assume that the length of the landslide is smaller than the wavelength under study, and thus, the temporal and spatial details of the source process cannot be resolved; accordingly, we employ a Bayesian full centroid moment tensor optimization (L1 norm, using Grond software[50]) using Rayleigh and Love waves between 0.01 and 0.03 Hz recorded by stations at distances reaching up to 500 km (Supplementary Figure 10). Time shift corrections were used to avoid bias in the centroid location. Green's function databases were calculated for regional Earth models (see https://greens-mill.pyrocko.org), following which a search was conducted for the centroid location, source duration, and full moment tensor components. Approximately 80,000 sampling iterations were used in the global optimization approach. Source parameter uncertainties were assessed from 100 bootstrap resampled realizations of the data set, which were explored in parallel. The centroid inversion estimated a location southwest of Anak Krakatau (Supplementary Figure 11). The ensemble of solutions indicates large uncertainties in the centroid location and depth. The trend of the uncertainty ellipsoid is in accordance with the station geometry with extension in the NW–SE direction. Therefore, we assume that the LP event was generated by the volcano sector collapse of Anak Krakatau. Finally, a stable

moment magnitude of Mw 5.3 was inferred from the full moment tensor inversion. The large difference between the $M_L$ and Mw reflects the relative deficiency in high frequencies of this event (compared with, e.g., tectonic earthquakes). The data fit obtained in the moment tensor optimization is lower than that obtained for a tectonic earthquake of similar size with the same setup. This finding suggests that a more complex source model may be needed to explain the observations, even for frequencies below 0.03 Hz.

**Drone photogrammetry**. Close-range drone photogrammetry data were acquired on 10 January 2019 by a GPS-controlled quadcopter drone (DJI Mavic Pro) equipped with a camera recording 1740 4 K images per flight minute and stabilized by a two-axis gimbal. In addition, we measured 14 reference points using a TanDEM-X digital elevation model (DEM) and the Sentinel-2 data acquired on 23 January 2019. In total, 450 high-quality images were selected and processed using the structure from motion and multiview stereo (SfM-MVS) approach[51–53]. This process yielded ~2.6 million points and allowed the generation of a DEM and a georeferenced photomosaic at a 1-m resolution. We compared the topography derived from the drone images with the topography from the TanDEM-X DEM and identified the locations of prominent morphological changes in an ArcGIS platform. As submarine areas are hidden to these methods, the volumetric loss and gain are minimum estimates (Supplementary Figure 15).

**Tsunami modeling**. We backtraced tsunami wavefronts starting from the positions of four coastal tide gauge stations: two along the Sumatran coast and two along the Java coast (Supplementary Figure 14). A tsunami travel time algorithm based on the classic Huygens–Fresnel principle[54] was applied to the 30-arcsecond Shuttle Radar Topography Mission Plus (SRTM30 +) model[55]. The expanding isochrones should theoretically intersect at one point, which corresponds to the position and initiation time of the tsunami source. The intersection point was indeed discovered at Anak Krakatau with an inferred origin time 2 mins after the seismically determined onset time of the landslide, which is within the range of the expected uncertainty for the backtracing procedure. The further reasons for these discrepancies include the limited quality and resolution of the bathymetric model, the finiteness of the source in space and time, the imprecise picking of the first arrivals at tide gauges, and possibly the nonnegligible wave dispersion typical of landslide sources.

**Infrasound**. Subaerially moving masses, such as landslides and volcanic explosions, release a significant amount of impulse energy into the air, which is radiated from the source as acoustic waves in the atmosphere. These waves propagate with average sound speeds of 250–350 m s$^{-1}$. We modeled infrasound propagation from Krakatau using the GeoAc raytracing suite[56] in the global 3D range-dependent propagation mode. Atmospheric parameters were derived from the forecast model of the European Centre for Medium-Range Weather Forecast (ECMWF—www.ecmwf.int), providing data for altitudes up to 75 km. We extended these data to higher altitudes (up to 150 km) with the empirical climatological wind and temperature models HWM14 and NRLMSIS00[57] (see the resulting effective velocities as the background colors of Supplementary Figure 12). The resulting infrasonic wavefield is displayed in the direction towards I06AU by the whole ray bundle and in map view by the locations (bounce points) where the sound waves emitted from Anak Krakatau reached the surface. We inspected the data from infrasound arrays operated by the International Monitoring System (IMS) of the Comprehensive Nuclear-Test-Ban Treaty Organization (CTBTO). The 22 December 2018 data from six stations (I04AU, I06AU, I07AU, I39PW, I40PG, and I52GB) were analyzed, but only the closest station, I06AU, situated 1150 km SW of Krakatau, recorded clear signals lasting for several hours that could be attributed to Krakatau. The modeling yielded two theoretical propagation paths (phases), one passing through the stratosphere (Is-phase) and one passing through the thermosphere (It-phase), corresponding to the two travel times of 3920 s (Is) and 4430 s (It), respectively. The Is and It phases of a sound wave emitted from Krakatau at 13:55:49 UTC (seismically determined origin time of the tremor signal) should arrive at I06AU at 15:01:09 UTC and 15:09:39 UTC, respectively. At I06AU, we detected clear signals that are consistent with the stratospheric phase (Is). We performed array analysis using the Obspy package in multiple frequency bands. The result for a 13-hour-long time interval using the 0.5–5 Hz frequency band and a 10 s window length with 5% overlap with a spectrogram from the best beam is shown in Supplementary Figure 8. The beam, which is consisted of high-frequency signals (0.5–5 Hz), starts to focus towards Krakatau at 7:45 UTC (6:39:20). The beam power shows an initial local maximum at ~13:00 UTC (11:54:20) and decreases afterwards. From 14:35 (13:29:40) to 14:40 UTC (13:34:40), the seismic beam is defocused, indicating a low beam power. During this time, the volcano seems to have been quiet. After 14:40 UTC (13:34:40), the volcanic activity increases slightly until 14:59 UTC (13:53:40), and a small impulsive signal is registered, followed by the main event at 15:01:09 (13:55:49). Times in parentheses are reduced by the infrasonic travel time of the stratospheric phase, derived from the modeling; in other words, these times indicate the origin time of the signal at Krakatau. After the main event, continuous energy was released from Krakatau (red dashed line in the spectrogram of Supplementary Figure 8c) but in a lower frequency band (0.1–0.7 Hz).

**SO₂ emission monitoring from Sentinel-5P data**. $SO_2$ emissions from Krakatau were recovered from the imaging spectrometer known as the Tropospheric Monitoring Instrument on board the Sentinel-5P satellite. Although the satellite was launched in 2017, the data have become progressively available only since late 2018. In this study, we analyze Level 2 (L2) offline (OFFL) data products downloaded from the Copernicus Sentinel-5P Pre-Operations Data Hub, where all products recorded after 5 December 2018 are available (https://scihub.copernicus.eu/news/News00440). $SO_2$ gas concentrations are provided at three different altitude ranges (as vertical column densities, VCDs): 0–1 km (the planetary boundary layer, PBL), 6.5–7.5 km (mid-troposphere), and 14.5–15.5 km (upper troposphere). $SO_2$ VCDs were first converted from mol m$^{-2}$ to Dobson units (DU) by a multiplication factor of 2241.15 (provided in the product metadata). The mass of $SO_2$ was then calculated[58] as follows:

$$MSO_2 = 0.0285 \times \sum_{i=0}^{n} A_i SO_{2i}$$

where $MSO_2$ is the mass of $SO_2$ (in tons) and $A_i$ and $SO_{2i}$ are the area ($7 \times 3.5$ km$^2$) and VCD (in DU) of each pixel, respectively. Pixels contaminated with $SO_2$ were isolated by creating a mask of DU > 1, to which a morphological filter (i.e., erosion + dilatation operators) was applied with a structuring element of $5 \times 5$ pixels. This allowed us to remove noisy pixels and keep only large pixel clusters, which were assumed to be of volcanic origin when these were located in the proximity of Krakatau. The $SO_2$ maps and time series provided here (Supplementary Figure 13) represent only the PBL.

## Data availability

The seismic data are available through EIDA and IRIS data centers (AU: www.iris.edu; GE: http://www.orfeus-eu.org/data/eida/), or are available on request, to be directed at BMKG (IA: https://www.bmkg.go.id/). MODIs data are available from the USGS hub at https://modis.gsfc.nasa.gov/data/, and Sentinel-2 satellite data available on the Copernicus Open Access Hub at https://scihub.copernicus.eu/, and Sentinel-5P data on the Copernicus Sentinel-5P Pre-Operations Data Hub. Any requests should be made to the first author. Infrasound data from the International Monitoring System (IMS) can be made available on request at the CTBTO (https://www.ctbto.org/specials/vdec/).

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

## Acknowledgements

This is a contribution to VOLCAPSE, a research project funded by the European Research Council under the European Union's H2020 Programme/ERC consolidator grant No. (ERC-CoG 646858), to the Research Network Geo.X, and to the Swedish Centre of Natural Hazards and Disaster Sciences (CNDS). F.M.S. is grateful to Stefano Serafin (University of Innsbruck) for providing great help to retrieve the atmospheric parameters for infrasound modeling. TerraSAR-X and TanDEM-X data are copyright of German Aerospace Agency (DLR) and were provided under the proposal IDs GEO1217, GEO1505, DEM_GEOL1196, and DEM_GEOL1670. We thank Mehdi Nikkhoo for contributing to the deformation modeling, and James Reynolds for sharing the drone videos. We thank Geoscience Australia, BMKG in Indonesia, and GEOFON (Germany) for making seismic data available.

## Author contributions

T.R.W. conducted the drone, Sentinel-1/2, and morphology analysis, M.H.H. and M.M. conducted the radar interferometry and deformation analysis. J.S., F.T. and F.M.S. analyzed the seismic data; T.D. and S.H. performed the moment tensor analysis; F.M.S. and P.G. realized the infrasound data analysis and modeling. A.B. performed the tsunami modeling; D.C., M.L. and F.M. investigated the satellite thermal data; S.V. processed and analyzed the Sentinel-5 data; R.T. contributed to tsunami and impact assessment; R.K. contributed to remote-sensing assessment, V.R.T. and N.K. analyzed and interpreted composition of the lava and ash samples. T.R.W. prepared the manuscript draft. All authors provided information and/or critical comments during manuscript preparation.
