## [Peer Review File · Nature Communications]

Reviewers' comments:

Reviewer #1 (Remarks to the Author):

Dear colleagues,

I'm impressed by the amount of data that you collected in a few weeks after the December 2018 collapse and tsunami of Anak Krakatau: MODIS satellite thermal data, SO₂ emissions (Sentinel 5), InSAR (Sentinel 1) and image analysis of the island perimeter (Sentinel 2), seismic data, infrasound, drone photogrammetry, etc. You particularly take advantage of Sentinel satellites data which are open to the community and allow all scientist to have a fast reaction in the case of such naturel disasters.

However, the manuscript falls far from the quality and innovation required for a publication in Nature Comunications. In fact it is more a report of the events, focused on the precursory signs and the event itself. And the "report" is very informative, but it is not what is expected for such high-impact journals. The dataset presented are all worth a more detailed analysis and line after line the reader realises that indeed there is more to say (e.g. InSAR, SO₂ data). Going deeper into the data, it's our job!

Other comments:

- More references could be added in the introduction (and when you decide to cite only one reference for a general topic, please avoid auto-citation... e.g. line 51).
- From the compositional analysis of ash sampled in July 2018 (i.e. 5 months before the collapse), you conclude that there is no deep magmatic change implied in the flank collapse. This statement would be valid only if geochemical data on the December 2018 juvenile magma was presented, which is the case.
- The 1m-resolution DEM built from the drone imagery is poorly presented and analysed (and apparently it covers only one part of the island): do you have an idea of the volume of new tephra that was deposited after the collapse? What's the accuracy of the ground control points extracted from the TanDEM and Sentinel 2 images?
- How do you distinguish between pure surface deformation and newly emplaced lavas on the southwestern flank of Anak on InSAR data? It's an important point of method that should be clarified.
- There's a mistake on subtitle of image 6 on figure S5.

To conclude, I'm sure that you have enough data to propose high-quality papers that will help better understanding what happens on December 22, 2018, but the way it is presented here is not satisfying.

Best regards

Reviewer #2 (Remarks to the Author):

This is a rather interesting paper, which provides evidence of previously unreported precursory volcano deformation affecting the rapidly growing cone of Anak Krakatau (Indonesia) during the months prior to its catastrophic collapse on 22 December 2018. It is based on a vast amount of data spanning a variety of disciplines that cover rock geochemistry, satellite remote sensing, and geophysics. One of the most fascinating aspects is that portions of the volcano were observed to subside even when the active cone was experiencing rapid growth at its summit. In hindsight, such significant volcano deformation would have been a precious warning sign, a variation on a common theme when thinking of the dramatic yet at the time not fully understood precursory deformation of Mount St Helens before its catastrophic collapse and eruption on 18 May 1980.

The paper is well written in virtually flawless English and is solid in its structure and discussion. However, I have come across a few sections that in my opinion merit some further albeit brief discussion, the most critical question being how much of Anak Krakatau was removed, in terms of volume and area lost to the island, by the 22 December 2018 collapse. Apart from that, the paper fully merits being published rapidly, because it is, in my view, a significant contribution to understanding this destructive and deadly event.

In the following I give more detailed comments.

Lines 54-55: It would be useful also to name the total number of volcanoes for which tsunamigenic events have been documented since 1600 AD, since in the following phrase it is said that 17 out of this number are located in Southeast Asia

Line 77: The number of victims was officially given as 426 in January 2019. If the figure given by the authors here is to be taken as more accurate, please provide source.

Line 205 and following: One of the main problems with understanding the full extent of the collapse lies in the fact that the first clear satellite images were available only on 28 December 2018, six days after the collapse, and intense post-collapse eruptive activity had already led to significant re-growth of the island, especially in terms of area. Most certainly the 28 December imagery does not represent the shape of the island as it was immediately after the collapse. However, this can be seen in a 22 December Sentinel-1 GRD image (acquired immediately after the event) that is shown in the supplementary Figure S3-1. Although that image is disturbed, presumably by the vigorous phreatomagmatic activity occurring at the time, it shows a much smaller island, with virtually the entire pre-collapse cone and most of the surrounding lava fields gone. I would therefore discuss the volume of the collapse and the area changes to the island with much caution and refrain from basing them on the 28 December 2018 imagery only. The collapse volume and area changes presented in the paper should therefore be given as minimum values, which in reality could have been up to twice those amounts; it is therefore also well possible that the entire area previously affected by subsidence was lost in the collapse, and not only 50% of it (lines 210-211).

Line 264 and following: Caution should be applied here – it cannot be excluded that there was a short-term, rapid change in the magma supply rate in the few hours before the catastrophic collapse; eyewitness accounts and photos taken on the very day of 22 December 2018 describe and show rather vigorous activity, with powerful Strombolian explosions and lava flows entering into the sea. A much smaller collapse of a portion of the New Southeast Crater cone of Mount Etna on 11 February 2014 had also been preceded by a short-lived increase in magma supply and eruption rate, accompanied by magma intrusion into the flank of the cone, which was interpreted by Andronico et al. (2018) as the principal trigger of the collapse. However, that cone had been already prone to instability due to its exceptionally rapid growth in the preceding three years, being positioned, like the growing cone of Anak Krakatau, on top of a steep slope. Unfortunately, no observations of deformation of the cone are available for the Etna 2014 event, making comparison with Krakatau 2018 difficult.

Reference: Andronico, D., Di Roberto, A., De Beni, E., Behncke, B., Bertagnini, A., Del Carlo, P., Pompilio, M. (2018) Pyroclastic density currents at Etna volcano, Italy: The 11 February 2014 case study. *Journal of Volcanology and Geothermal Research* 357: 92-105.

Catania, 21 April 2019

Boris Behncke

Istituto Nazionale di Geofisica e Vulcanologia

Reviewer #3 (Remarks to the Author):

The authors present an intriguing analysis of processes leading up to the unprecedented tsunami that hit Indonesia in December 2018 as a result of a sector collapse at Anak Krakatau volcano in the Sunda Strait. The research underpinning the paper is sound and reported using a narrative that is transparent and tractable.

Obviously, a huge effort has been made to bring together a large multidisciplinary dataset and interpret it jointly and simultaneously. This is no mean feat. Each data stream could have been evaluated in isolation to produce a paper. The strength of the current ms is that it really aims to provide a holistic view of the anatomy of surface processes at Anak Krakatau in the lead-up to the catastrophic tsunamigenic collapse. This provides the reader with an exemplar analysis that is cogent and convincing. As the author's rightly state, neither of the observations would have yielded much hindsight wisdom if analysed in isolation. In unison, however, the data picture a perfect storm of magma-tectonic processes which culminated in one of the deadliest volcano collapses in the recent past.

The paper should appeal to a broad readership including expert in the respective fields of remote sensing of natural hazards, geology, geodynamics, geophysics and natural risk governance as well as readers with a keen interest in how our planet works.

The paper has all the pedigree for publication in Nature Communications, but there are a couple of points that ought to be addressed by the authors:

- 1) Although written to an acceptable standard, the narrative deserves a detailed overhaul by a native English speaker. There are several instances of awkward phrasing and ambiguous formulations. The narrative can be made much crisper, succinct and more cogent. I have not attempted to correct the English.

2) Consideration of uncertainties -- Some parameter values derived from the analysis report uncertainties while others don't. The authors should ensure proper uncertainty estimation and error propagation throughout. As an example, I point to the abstract which reports a collapse volume without providing error bounds.

3) Abstract-- I do not think the abstract is written in a compelling way. It follows a pedestrian-style approach and should be rewritten to better report the gist of the study including numerical values of the observations. Being aware of length restrictions, I recommend the authors use a more succinct narrative from the outset to make room for vital information. BTW the reader does not gain a clear idea when the collapse and tsunami actually happened, as the sentence containing the date of Dec 22 is framed ambiguously.

4) The paper is very much focussed on reporting observations and drawing conclusions from them. Although perhaps not possible for all data streams, the paper could be strengthened by investing some time to provide more cogent insights into the driving processes during the lead-up to the collapse. For example, the deformation data points towards ground subsidence and outward movement of the western flank. What are the dominant surface and subsurface process driving this movement? Can the authors provide a quantitative model to explain the observations? Seismic and infrasound data are used to derive source models, why do the authors not do the same for the deformation data? At the moment it is unclear what role surface loading vs subsurface loading play in the preparatory phase. Lava flow loading is mostly in the south, so why did the southern flank not collapse? Getting some quantitative information on the role of different contributors to the deformation field would strengthen the paper. For example, intrusive volumes can be linked to eruptive volumes etc.

5) Figures:

Fig 1 : a) worth providing a lat/long marker? c) The area undergoing collapse between Dec 18 and Jan 19 should be better highlighted. I cannot see a white line in the figure as described in the caption. panels d), f) and g) should provide x-labels (2018|2019)

Fig 2: "The catastrophic event". Suggest to avoid sensationalism despite the dramatic impact. c and d) I suspect normalized amplitude is reported? Should be made clear since no unit is given.

Fig 3. a) make clear that these are time series images taken on three different days. Pls use consistent date format in text, captions and figures. This gets very confusing otherwise. b) I'm not sure I fully understand the caption given what is portrayed. Please highlight the structures that you interpret faults and folds. To my eye given the limited insight from the image provided I can make out some valley incisions and perhaps a more pronounced dark gray curved lineament similar to an incised stream bed in the zoom-in.

You highlight I, II and III but do not provide information in the caption why you think these are anomalous. I guess you're after the steps/offsets which are more obvious in II and III?

Fig 4: Orientations of the cross-sections are missing. the use of colors and symbols need to be fully explained. The caption is not a lone-standing item at the moment that provides all information for the understanding of the figure.

Please find below the reviewers' comments followed by **our response in bold**:

Reviewer #1 (Remarks to the Author):

Dear colleagues,

I'm impressed by the amount of data that you collected in a few weeks after the December 2018 collapse and tsunami of Anak Krakatau: MODIS satellite thermal data, SO₂ emissions (Sentinel 5), InSAR (Sentinel 1) and image analysis of the island perimeter (Sentinel 2), seismic data, infrasound, drone photogrammetry, etc. You particularly take advantage of Sentinel satellites data which are open to the community and allow all scientist to have a fast reaction in the case of such naturel disasters.

However, the manuscript falls far from the quality and innovation required for a publication in Nature Communications. In fact it is more a report of the events, focused on the precursory signs and the event itself. And the "report" is very informative, but it is not what is expected for such high-impact journals. The dataset presented are all worth a more detailed analysis and line after line the reader realises that indeed there is more to say (e.g. InSAR, SO₂ data). Going deeper into the data, it's our job!

Reply: We agree with the reviewer and have now put more emphasis into detailed analysis of our presented data. In the revised version, we now go much deeper into the data, clarify the meaning of the increased thermal flux (higher magma productivity), the meaning of continued deformation (which we model now through inversion approaches) and the comparison of the different data streams. Moreover, we have added and improved the description of the InSAR derived decollement slip, unloading of the hydrothermal system and SO₂ degassing. Full details of the changes made are given in the replies to the related comments below.

Other comments:

- More references could be added in the introduction (and when you decide to cite only one reference for a general topic, please avoid auto-citation... e.g. line 51).

Reply: We carefully checked the referencing. We firstly fixed a mistake (citation of Paris et al., 2014 instead of Paris et al. 2015) and added new relevant references (Agustan et al., 2012; Chaussard and Amelung, 2012). We now also reduced auto-citations where possible.

In line 51 we have now added a reference to McGuire, W.J., 1996. Volcano instability: a review of contemporary themes. Geological Society, London, Special Publications, 110: 1-23.

In the lines 64 and afterwards, we improved the referencing by adding a description of the different source types discussed in the literature.

- From the compositional analysis of ash sampled in July 2018 (i.e. 5 months before the collapse), you conclude that there is no deep magmatic change implied in the flank collapse. This statement would be valid only if geochemical data on the December 2018 juvenile magma was presented, which is the case.

Reply: We appreciate this comment and clarified the text. In fact the intention of the sampling was misunderstood, which is why we now rephrased this part. The sample was taken during the time of increased eruption rate (as we identified in the Modis data, see figure 1). In fact, the sample was taken during one of the largest eruptions in the past 20 years, and during the climax of magmatic activity in 2018. The analysis of the sample

reveals that the composition is not different to the ones sampled in the years before. Therefore our statement is that there is no deep magmatic change associated with the increased eruption rate. However, whether the 22 Dec 2018 sector collapse is causing a deep change at Anak Krakatau has to be investigated in future studies, as this phenomenon was observed for volcano collapses elsewhere. We made appropriate changes in Lines L 270 and following.

- The 1m-resolution DEM built from the drone imagery is poorly presented and analysed (and apparently it covers only one part of the island): do you have an idea of the volume of new tephra that was deposited after the collapse? What's the accuracy of the ground control points extracted from the TanDEM and Sentinel 2 images?

Reply: We agree that this warrants further description. We now clarified the resolution of the satellite data in the methods section, and specify in the supplementary material S9 that we do not correlate to these few points, but rather co-align the entire point clouds using CloudCompare. Changes are made in the methods section L442- 458 and S9.

It is correct that the drone data only covers part of the island. In our supplement S9 we now describe: "...the reconstructed point cloud quality is highest on the southeastern sector of Anak Krakatau." The drone flights were realized from a fishing vessel outside the 5 km exclusion zone, therefore the northwestern side of the island was beyond the reach of the drone". We have explained this limitation in the revised supplement S9. The volume of new tephra is indeed difficult to quantify from the data available, which is why we discuss this point with much greater care. We now state that an accurate measurement of the volume would need to have a precise DEM immediately prior to the collapse, and a precise DEM immediately after the collapse, plus an exact knowledge of the submarine proportion as well. We discuss these challenges in the new S10 subsection on "Material loss and gain".

The accuracy of a TanDEM-X dataset is provided in Wessel et al. (2018), to whom we now refer (Wessel et al., 2018). We made appropriate changes in the S9 description, where we changed the formulation to: "Direct Ground Control Points (GCPs) could not be acquired due to the exclusion zone and hazardous access to the volcano. Therefore we now follow the strategy described in (Muller et al., 2017) by using the internal GPS of the drone for coarse location, comparison to the GCPs, then co-alignment of the three-dimensional point cloud to an independent dataset of a known quality. A cloud comparison to TanDEM-X topographic data was realized using the point pair-picking registration in the CloudCompare software (version 2.9.1) [GPL software]. This yielded a reprojection error of 8.9 pixels. This relatively large reprojection error results from the viewing geometry, as flights could only be realized from a large distance and from the southern and southeastern side. The vertical mean error of TanDEM-X data is smaller than 0.2 m, and has a Root Mean Square Error (RMSE) smaller 1.4 m (Wessel et al., 2018)."

- How do you distinguish between pure surface deformation and newly emplaced lavas on the southwestern flank of Anak on InSAR data? It's an important point of method that should be clarified.

Reply: We agree that this was in need of more detail. Newly emplaced lavas are seen by coherence loss (Fig S4-5) and in thermal images (Fig. S1-2). Deformation, in turn, is shown only at those pixels that remain coherent in the InSAR time series.

Changes made in L148-153, L154, L160-163 of the main text, and L385-386 and L390-393 of the methods section.

- There's a mistake on subtitle of image 6 on figure S5.

Reply: Accepted and changes made.

To conclude, I'm sure that you have enough data to propose high-quality papers that will help better understanding what happens on December 22, 2018, but the way it is presented here is not satisfying.

Reply: We appreciate the comments by reviewer 1 and have addressed all points raised. The input of reviewer 1 has helped us to substantially improve the manuscript.

Best regards

Reviewer #2 (Remarks to the Author):

This is a rather interesting paper, which provides evidence of previously unreported precursory volcano deformation affecting the rapidly growing cone of Anak Krakatau (Indonesia) during the months prior to its catastrophic collapse on 22 December 2018. It is based on a vast amount of data spanning a variety of disciplines that cover rock geochemistry, satellite remote sensing, and geophysics. One of the most fascinating aspects is that portions of the volcano were observed to subside even when the active cone was experiencing rapid growth at its summit. In hindsight, such significant volcano deformation would have been a precious warning sign, a variation on a common theme when thinking of the dramatic yet at the time not fully understood precursory deformation of Mount St Helens before its catastrophic collapse and eruption on 18 May 1980.

Reply: Good point. We have now strengthened reference to the Mount St Helens case. The reviewer has helped us to clarify the importance of our findings. Changes made in L 279-282.

The paper is well written in virtually flawless English and is solid in its structure and discussion. However, I have come across a few sections that in my opinion merit some further albeit brief discussion, the most critical question being how much of Anak Krakatau was removed, in terms of volume and area lost to the island, by the 22 December 2018 collapse. Apart from that, the paper fully merits being published rapidly, because it is, in my view, a significant contribution to understanding this destructive and deadly event.

Reply: We are glad to read this opinion by reviewer 2! We have now responded and added details to the volume and area loss.

In the following I give more detailed comments.

Lines 54-55: It would be useful also to name the total number of volcanoes for which tsunamigenic events have been documented since 1600 AD, since in the following phrase it is said that 17 out of this number are located in Southeast Asia

Reply: We appreciate this comment. We added the missing information and also included the reference to the Tsunami Database. The 130 events are a described minimum number in Paris et al. (2015) and listed at 80 different volcanoes as given by the NGDC/WDS Global Historical Tsunami Database of the National Geophysical Data Center. As correctly pointed out by the reviewer, the number of volcanoes is different, as some of the events occurred at the same volcano, for instance twice on Hawaii (1877, 2018), twice at Kick em Jenny (1939 and 1965), and four times at Krakatau (1883, 1928,

1930, 2018). Not all of tsunami sources can be clearly located, however, but we can identify at least 80 distinct volcano locations for the period since 1600 AD. This is now outlined in the revised manuscript.

References corrected:

Paris, R., Switzer, A.D., Belousova, M., Belousov, A., Ontowirjo, B., Whelley, P.L. and Ulvrova, M., 2014. Volcanic tsunami: a review of source mechanisms, past events and hazards in Southeast Asia (Indonesia, Philippines, Papua New Guinea). *Nat Hazards*, 70(1): 447-470.

NGDC/WDS, 2019. Global Historical Tsunami Database. NOAA National Centers for Environmental Information. doi:10.7289/V5PN93H7
[<http://www.ngdc.noaa.gov/hazard/hazards.shtml> access June 2019]

Line 77: The number of victims was officially given as 426 in January 2019. If the figure given by the authors here is to taken as more accurate, please provide source.

Reply: Accepted and changes made. UBNPB reported 437 fatalities, 16 missing, 14,059 injured and 33,719 displaced and we rewrote this sentence accordingly.

Data source: 2018 Press Release - Tsunami events on the West Coast of Banten were not triggered by earthquakes. 22 December 2018. Summarized in National Geophysical Data Center—NGDC doi:10.7289/V5PN93H7.

Line 205 and following: One of the main problems with understanding the full extent of the collapse lies in the fact that the first clear satellite images were available only on 28 December 2018, six days after the collapse, and intense post-collapse eruptive activity had already led to significant re-growth of the island, especially in terms of area. Most certainly the 28 December imagery does not represent the shape of the island as it was immediately after the collapse. **We agree, this is correct !** However, this can be seen in a 22 December Sentinel-1 GRD image (acquired immediately after the event) that is shown in the supplementary Figure S3-1. Although that image is disturbed, presumably by the vigorous phreatomagmatic activity occurring at the time, it shows a much smaller island, with virtually the entire pre-collapse cone and most of the surrounding lava fields gone. I would therefore discuss the volume of the collapse and the area changes to the island with much caution and refrain from basing them on the 28 December 2018 imagery only.

Reply: We appreciate this comment very much. Indeed, we reexamined the available dataset so that in the revised version we now carefully consider this issue. We assessed the 22 Dec 2018 GRD radar image, and conclude that a significant proportion of the southeastern sector is affected by radar path artifacts in this particular acquisition, an effect that has been described in the scientific literature before: Volcanic gas plumes may lead to clear radar path effects as described in Bredemeyer et al. (2018). We note that also ash clouds have severe effects such as described in Meyer et al. (2014).

Therefore we are indeed careful when interpreting the blurred southwestern part of the images from 22 Dec and 25 Dec 2018, and have added this detail to the text in L 233-237 and in caption S3-1. To further convince the reviewer we provide a close up of the relevant GRD images.

References added:

Bredemeyer, S., Ulmer, F. G., Hansteen, T. H. & Walter, T. R. Radar Path Delay Effects in Volcanic Gas Plumes: The Case of Lascar Volcano, Northern Chile. *Remote Sens-Basel* 10, doi:10.3390/rs10101514 (2018).

Meyer, F. J. et al. Integrating SAR and derived products into operational volcano monitoring and decision support systems. *Isprs J Photogramm* 100, 106-117, doi:10.1016/j.isprsjprs.2014.05.009 (2015).

Figure S3-2. Enlarged Sentinel 1 GRD image catalogue before the collapse (upper left image) and on the day of the collapse (the collapse was at 13:55 UTC, the GRD image was acquired 8.5 hours afterwards at 22:33 UTC) and afterwards. Arising from higher radar reflectivity, steep cliffs of landslide amphitheater can be partly identified in the 22 Dec 2018 image (dashed black line). In other parts this image is presenting radar signal artifacts arising from the eruption plume. Note the old coastline in the northeast. On 25

Dec 2018, the southwest sector is still blurred, and again interpreted by us to be a consequence of the eruption plume. We note that on the northern coastline now appears shifted due to new volcano products. The image on 27 December is the first which reveals the complete amphitheater geometry (partially infilled in the northwest) and the new coastline (shifted by up to 260 m) in the northeast. Dashed black lines in second and fourth image represent approximate outlines of satellite headwalls, which might indicate two outlines, enclosing approximately 0.63km² and 0.84km², which is 44% and 58% of the deformation area identified in InSAR time series, respectively. We have added this detail in the text L 233-237 and in supplement S3.

The collapse volume and area changes presented in the paper should therefore be given as minimum values, which in reality could have been up to twice those amounts; it is therefore also well possible that the entire area previously affected by subsidence was lost in the collapse, and not only 50% of it (lines 210-211).

Reply: We re-estimated this aspect. The difference of the two digital elevation models suggest a volume change as we correctly stated. However, we have now improved the manuscript by detailing the area assessment (see above and S3-2) that this is combining volume loss (by collapse and erosion) and volume gain (by tephra deposition), both of which are now discussed in the revised manuscript. Concerning the area of the collapse and the area of the pre-collapse deformation, we added further detail by inserting the collapse outline in the InSAR map, showing that the actual deformation area was much larger than the collapsed area (assessed to be between 44% and 58% of the deformation area). This variation is based on the GRD radar image taken on 22 Dec 2018 and 25 Dec 2018, which both are not to be overinterpreted due to artifacts, however. We now rephrased lines L233-252 and S3 section accordingly, and have removed unsupported statements.

Line 264 and following: Caution should be applied here – it cannot be excluded that there was a short-term, rapid change in the magma supply rate in the few hours before the catastrophic collapse; eyewitness accounts and photos taken on the very day of 22 December 2018 describe and show rather vigorous activity, with powerful Strombolian explosions and lava flows entering into the sea.

Reply: Accepted and rewritten. We now write “Analysis of the infrasound array data recorded at station I06AU of the IMS of the CTBO (12.1491°S, 96.8221°E, distance from Krakatau: 1158.8 km) in the hours before the sector collapse reveals that Anak Krakatau was already in an elevated state. We note that approximately one hour before the sector collapse, the activity is markedly reduced, even falling to levels insufficient to form a coherent beam for approximately 15 minutes (from ~14:35 arrival time)”. This is in line with eyewitness accounts, which we describe as follows: “Eye-witness accounts support these findings: the activity increased to a subjective peak 2-3 hours before the sector collapse at 11:30 UTC, and when new materials deposited a strong red glow formed, with the eruption turning more ash rich by 12:00 UTC. During the half hour before the sector collapse (13:30-14:00 UTC), the eruption sounds stopped”. See S7.

We now also clarify the possibility of short term pulses that might have occurred in the main text.

A much smaller collapse of a portion of the New Southeast Crater cone of Mount Etna on 11 February 2014 had also been preceded by a short-lived increase in magma supply and eruption rate, accompanied by magma intrusion into the flank of the cone, which was interpreted by Andronico et al. (2018) as the principal trigger of the collapse. However, that

cone had been already prone to instability due to its exceptionally rapid growth in the preceding three years, being positioned, like the growing cone of Anak Krakatau, on top of a steep slope. Unfortunately, no observations of deformation of the cone are available for the Etna 2014 event, making comparison with Krakatau 2018 difficult.

Reference: Andronico, D., Di Roberto, A., De Beni, E., Behncke, B., Bertagnini, A., Del Carlo, P., Pompilio, M. (2018) Pyroclastic density currents at Etna volcano, Italy: The 11 February 2014 case study. *Journal of Volcanology and Geothermal Research* 357: 92-105.

Reply: Accepted. Interesting case to compare our data with. We have now added this reference in L 316-218.

Reviewer #3 (Remarks to the Author):

The authors present an intriguing analysis of processes leading up to the unprecedented tsunami that hit Indonesia in December 2018 as a result of a sector collapse at Anak Krakatau volcano in the Sunda Strait. The research underpinning the paper is sound and reported using a narrative that is transparent and tractable.

Obviously, a huge effort has been made to bring together a large multidisciplinary dataset and interpret it jointly and simultaneously. This is no mean feat. Each data stream could have been evaluated in isolation to produce a paper. The strength of the current ms is that it really aims to provide a holistic view of the anatomy of surface processes at Anak Krakatau in the lead-up to the catastrophic tsunamigenic collapse.

Reply: Thank you for the supportive comments on our work.

This provides the reader with an exemplar analysis that is cogent and convincing. As the author's rightly state, neither of the observations would have yielded much hindsight wisdom if analysed in isolation. In unison, however, the data picture a perfect storm of magma-tectonic processes which culminated in one of the deadliest volcano collapses in the recent past.

Reply: We like the expression "storm of magma-tectonic processes", and have now adopted it at the end in our revised manuscript.

The paper should appeal to a broad readership including expert in the respective fields of remote sensing of natural hazards, geology, geodynamics, geophysics and natural risk governance as well as readers with a keen interest in how our planet works.

Reply: We thank reviewer 3 for the supportive words.

The paper has all the pedigree for publication in *Nature Communications*, but there are a couple of points that ought to be addressed by the authors:

1) Although written to an acceptable standard, the narrative deserves a detailed overhaul by a native English speaker. There are several instances of awkward phrasing and ambiguous formulations. The narrative can be made much crisper, succinct and more cogent. I have not attempted to correct the English.

Reply: Accepted. We have now had the manuscript read by a native speaker.

2) Consideration of uncertainties -- Some parameter values derived from the analysis report

uncertainties while others don't. The authors should ensure proper uncertainty estimation and error propagation throughout. As an example, I point to the abstract which reports a collapse volume without providing error bounds.

Reply: We agree. Uncertainties are provided for all data now, and have been addressed in the revised description of the collapse volume.

To specifically address the issue of the collapse volume, we carefully re-examined the two digital elevation models, which indeed combine volume loss (by collapse and erosion) and volume gain (by tephra deposition), both of which are now discussed in detail in the revised manuscript. As suggested by reviewer 2 we now use the Sentinel-1 data to constrain this aspect. Sentinel-1 yields the island perimeter, showing a shift of 120 up to 260 m of the northeastern coastline associated with tephra deposition (see also figure presented in this response letter, and figure S3-1 in the supplementary material). We calculated a volume again for the northeastern island sector of $3.17 \times 10^7 \text{ m}^3$, which was added between the GRD images acquired on 22 and 27 of December 2018.

Therefore, the volume gain over the island may be as large as $1 \times 10^8 \text{ m}^3$, and the total volume change derived from the two DEMs (before the collapse, and from January drone data) is underestimated as noted by reviewer 2. We note, however, that the exact volume can not be assessed from the data available, as we lack the submarine part. Since the submarine part is completely missing in this assessment, the total volume of the collapsed flank may be even larger than the corrected values described, as we now thoroughly discuss in the main text discussion chapter and in the supplementary chapter S10. (Material loss and gain).

3) Abstract-- I do not think the abstract is written in a compelling way. It follows a pedestrian-style approach and should be rewritten to better report the gist of the study including numerical values of the observations. Being aware of length restrictions, I recommend the authors use a more succinct narrative from the outset to make room for vital information. BTW the reader does not gain a clear idea when the collapse and tsunami actually happened, as the sentence containing the date of Dec 22 is framed ambiguously.

Reply: We appreciate this comment and have rewritten the abstract accordingly (see above). We also added the exact timing of the collapse as suggested. We hope the reviewer 3 will find the abstract is more compelling now.

4) The paper is very much focussed on reporting observations and drawing conclusions from them. Although perhaps not possible for all data streams, the paper could be strengthened by investing some time to provide more cogent insights into the driving processes during the lead-up to the collapse. For example, the deformation data points towards ground subsidence and outward movement of the western flank. What are the dominant surface and subsurface process driving this movement?

Reply: We now address this comment in the various parts of the paper.

Firstly, we better describe the possible process coupling between the observations (e.g. SO₂ peak during the timing of eruption deposition), added more detail on volume loss and gain, and describe the interpretation and modelling with greater care. As we now also argue that deeper processes can be inferred from seismicity, which suggests a steeply SW dipping failure nodal plane, possibly representing the landslide decollement. This is also inferred from the (new) inversion of InSAR data, suggesting a SW dipping dislocation plane. Therefore, as the InSAR data is pre-collapse, and the seismic data is syn-collapse, we conclude that the landslide decollement has evolved already before the collapse. Changes were made in L 298 and thereafter. In addition, we now also discuss possible driving mechanisms, such as the added load caused by increasing eruption

deposits on the surface in 2018, and the possible increase of pore pressure, and we now refer to relevant literature. We now also compare our results to other volcanoes, where an increase in the magma flux rate prior to collapse was also observed, similar to our findings (cf. Andronico et al., 2018).

Can the authors provide a quantitative model to explain the observations? Seismic and infrasound data are used to derive source models, why do the authors not do the same for the deformation data?

Reply: Yes, we agree. We now added a new set of models to invert for the source mechanism of the InSAR displacement field and compare these observations to seismic data. In fact, the results are very interesting, as they allow comparison with results from the seismic model, e.g. strike and dip, slumping direction and geometry of the headwall. This is now stated in the revised version of the manuscript (L300 and following), in the methods section and in the supplementary material (S 4).

(a) InSAR Data

(b) Dislocation model

(c) Data - Model

Figure showing (a) unwrapped mean line of sight (LOS) velocity data for the investigated tracks and viewing geometries, (b) the synthetic LOS displacements created from inversion dislocation models, and (c) the residual created by calculating the difference between the data and the model. The parameters of a best fitting decollement plane, simulated as a rectangular dislocation, are 35° dip, and 163° strike, with a -90.5° rake, and 3.36m slip.

At the moment it is unclear what role surface loading vs subsurface loading play in the preparatory phase. Lava flow loading is mostly in the south, so why did the southern flank not collapse? Getting some quantitative information on the role of different contributors to the deformation field would strengthen the paper. For example, intrusive volumes can be linked to eruptive volumes etc.

Reply: We agree that in the earlier version, this was not described in the necessary detail. Based on the remote sensing data we are able to derive the volume of new lava flows deposited mostly in the south, where 11 significant eruptive pulses are recorded (see also figure 1). According to Sentinel 2 data, which is higher resolution than Modis, during the pre-collapse period in 2018 at least 0.85 km² of the island were covered by abundant hot ejecta and lava flows that reached the sea several times and added about 0.1 km² to the island. Lava volume calculation, derived from thermal data (see methods), indicates that the 2018 eruption produced 2.5×10^7 m³ of lava. We note that the uncertainty is large, estimated to $\pm 1.27 \times 10^7$ m³ of lava. The Mean Output Rate (MOR) is calculated to 1.7 ± 0.8 m³ s⁻¹. Therefore, in terms of the volume, about 1/4 of the estimated collapse volume (1×10^8 m³) was deposited as fresh material prior on the volcano flanks during the 6 months prior to the eruption (this we clarified in L 134 and following). The reason why the southern tip did not collapse can not be established at present, but we see that the part that was moving vertical and sliding to the SW (this we clarified in L155 and in 166-167).

The intrusive volume can not be constrained. In fact, the observed deformation data can be explained by a decollement fault without any opening (see above figure and description), implying that intrusive volume is minor. We note that largest residuals occur at locations of lava flow deposition prior to the collapse, associated with cooling and contraction, rather than flank movement. We now state this in the revised manuscript (L 292-295).

5) Figures:

Fig 1 : a) worth providing a lat/long marker? c) The area undergoing collapse between Dec 18 and Jan 19 should be better highlighted. I cannot see a white line in the figure as described in the caption. panels d), f) and g) should provide x-labels (2018|2019)

Reply: Accepted and changes made. We have now added a lat/lon marker as suggested for figure 1a). We have added the collapse amphitheater outline as suggested for figure 1c) and also updated the caption accordingly. We have now added the x-labels 2018/2019 in figures 1d), f), and g).

Fig 2: "The catastrophic event". Suggest to avoid sensationalism despite the dramatic impact. c and d) I suspect normalized amplitude is reported? Should be made clear since no unit is given.

Reply: Accepted. We have removed "the catastrophic event" from the caption and write "Seismic and infrasound recordings of volcano sector collapse". Amplitude is normalized, correct, and we added this to the caption.

Fig 3. a) make clear that these are time series images taken on three different days. Pls use consistent date format in text, captions and figures. This gets very confusing otherwise. b) I'm not sure I fully understand the caption given what is portrayed. Please highlight the structures that you interpret faults and folds. To my eye given the limited insight from the image provided I can make out some valley incisions and perhaps a more pronounced dark gray curved lineament similar to an incised stream bed in the zoom-in.

You highlight I, II and III but do not provide information in the caption why you think these are anomalous. I guess you're after the steps/offsets which are more obvious in II and III?

Reply: Accepted and changes made. We now use consistent date format. We agree that the interpreted faulting or folding architecture was weak and therefore removed it from

the paper. In the supplementary figure S4-4 we state that these jumps may arise from faulting or from new deposited eruption material.

Fig 4: Orientations of the cross-sections are missing. the use of colors and symbols need to be fully explained. The caption is not a lone-standing item at the moment that provides all information for the understanding of the figure.

Reply: Accepted. We now added the orientation of the cross-sections (NE-SW). The symbols and colors are now explained in a legend. We describe the seismic event before the collapse, the infrasound and low frequency (LF) source, the decompression of the interior, and new deposits. The caption was now rewritten.

We hope the reviewers find our responses adequate and look forward to their recommendations.

- Agustan, Kimata, F., Pamitro, Y.E. and Abidin, H.Z., 2012. Understanding the 2007-2008 eruption of Anak Krakatau Volcano by combining remote sensing technique and seismic data. *Int J Appl Earth Obs*, 14(1): 73-82.
- Chaussard, E. and Amelung, F., 2012. Precursory inflation of shallow magma reservoirs at west Sunda volcanoes detected by InSAR. *Geophys Res Lett*, 39: L21311.
- Muller, D., Walter, T.R., Schopa, A., Witt, T., Steinke, B., Gudmundsson, M.T. and Durig, T., 2017. High-Resolution Digital Elevation Modeling from TLS and UAV Campaign Reveals Structural Complexity at the 2014/2015 Holuhraun Eruption Site, Iceland. *Front Earth Sci*, 5.
- Wessel, B., Huber, M., Wohlfart, C., Marschalk, U., Kosmann, D. and Roth, A., 2018. Accuracy assessment of the global TanDEM-X Digital Elevation Model with GPS data. *Isprs J Photogramm*, 139: 171-182.

REVIEWERS' COMMENTS:

Reviewer #2 (Remarks to the Author):

I found the revised manuscript very satisfying, having considered virtually every point raised by all reviewers and gone into some depth as for the issues that I raised on the original manuscript. So my answer is yes, I am indeed happy with the revisions made by the authors.

Reviewer #3 (Remarks to the Author):

The authors have provided a revised version of their manuscript in which they have made a number of changes as a result of reviewers' comments. In my view the ms has been substantially improved and I particularly welcome the addition of more quantitative analysis of the precursory signals. I still have a few points to raise regarding the revised ms and they should be straightforward to implement. Most of them relate to the the writing, which I still think can be improved. While the abstract has been completely overhauled there are still weaknesses that should be looked at. The abstract is one of the main selling points of the story and the authors should perhaps be more careful at getting the gist of their findings across. I have attempted a possible edit below.

1. Suggested edit for abstract:

Flank instability and sector collapses are common on volcanic islands and pose a major threat. On 22 December, 2018 a sector collapse occurred at Anak Krakatau volcano in the Sunda Strait triggering a deadly tsunami. Here, we use multi-parametric ground-based and space-borne data to show that the volcano exhibited an elevated state of activity over several months prior to the collapse including precursory thermal anomalies, an increase in the island's surface area, and a gradual seaward motion of its southwestern flank. Two minutes after a small earthquake, seismic signals characteristic of a rockslide-avalanche indicate a collapse of the volcano's western flank towards the southwest at 13:55 UTC. This sector collapse decapitated the cone-shaped edifice and triggered a tsunami that caused 346 fatalities on neighbouring islands. The collapse was followed by intense explosions and sustained degassing. We discuss the nature of the precursory processes underpinning the collapse and which culminated in a complex hazard cascade, with important implications for early detection of potential flank instability at other volcanoes.

2. While I appreciate the use of the term "perfect storm" coined in my previous review, its use in the revised version does not make much sense. I suggest to edit lines 326 onwards: "It appears that a perfect storm of magma-tectonic processes at Anak Krakatau culminated in the 22 Dec 2018 tsunami disaster. Leading up to the event, different sensors and methods measured distinct anomalous behaviour, which in hindsight can be deemed 'precursory'. However, at the time and in isolation none of the parameters including thermal anomalies, flank motion, anomalous degassing, seismicity, and infrasound, were sufficiently conclusive to shed light on the events that were about to unfold."

3. It would only be fair to cite :

Rockfall-Avalanche and Rockslide-Avalanche Deposits at Sawtooth Ridge, Montana
M. R MUDGE
GSA Bulletin (1965) 76 (9): 1003-1014.

where definitions are given in relation to event described in the manuscript. Note the hyphen (-) in

the term Rockslide-Avalanche compared to Rockslide Avalanche used in the ms. I suggest you stick with the original term and spelling coined by Mudge.

4. The authors refer to the mass movement as a "landslide", "rockslide" or "rockslide-avalanche". Please do not use these terms loosely as they entail different source processes and materials undergoing failure. Be sure you use the correct term and stick to it throughout the ms incl supp mats.

L65: delete "potential"

L341: delete "behaviour"; in fact the entire sentence is illogical. Is it not because these events are rare that multiparameter insights such as those reported here yield vital information on precursory processes?

Congratulations on a fine paper. J Gottsmann

REVIEWERS' COMMENTS:

Reviewer #2 (Remarks to the Author):

I found the revised manuscript very satisfying, having considered virtually every point raised by all reviewers and gone into some depth as for the issues that I raised on the original manuscript. So my answer is yes, I am indeed happy with the revisions made by the authors.

Reply: We are glad to read this.

Reviewer #3 (Remarks to the Author):

The authors have provided a revised version of their manuscript in which they have made a number of changes as a result of reviewers' comments. In my view the ms has been substantially improved and I particularly welcome the addition of more quantitative analysis of the precursory signals. I still have a few points to raise regarding the revised ms and they should be straightforward to implement. Most of them relate to the the writing, which I still think can be improved. While the abstract has been completely overhauled there are still weaknesses that should be looked at. The abstract is one of the main selling points of the story and the authors should perhaps be more careful at getting the gist of their findings across. I have attempted a possible edit below.

Reply: We are glad to read this and have considered all suggestions by reviewer #3 as detailed below.

1. Suggested edit for abstract:

Flank instability and sector collapses are common on volcanic islands and pose a major threat. On 22 December, 2018 a sector collapse occurred at Anak Krakatau volcano in the Sunda Strait triggering a deadly tsunami. Here, we use multi-parametric ground-based and space-borne data to show that the volcano exhibited an elevated state of activity over several months prior to the collapse including precursory thermal anomalies, an increase in the island's surface area, and a gradual seaward motion of its southwestern flank. Two minutes after a small earthquake, seismic signals characteristic of a rockslide-avalanche indicate a collapse of the volcano's western flank towards the southwest at 13:55 UTC. This sector collapse decapitated the cone-shaped edifice and triggered a tsunami that caused 346 fatalities on neighbouring islands. The collapse was followed by intense explosions and sustained degassing. We discuss the nature of the precursory processes underpinning the collapse and which culminated in a complex hazard cascade, with important implications for early detection of potential flank instability at other volcanoes.

Reply: accepted and changes made. We also let the text proofread by a native speaker, who helped further clarifying remaining issues.

2. While I appreciate the use of the term "perfect storm" coined in my previous review, its use in the revised version does not make much sense. I suggest to edit lines 326 onwards: "It appears that a perfect storm of magma-tectonic processes at Anak Krakatau culminated in the 22 Dec 2018 tsunami disaster. Leading up to the event, different sensors and methods measured distinct anomalous behaviour, which in hindsight can be deemed 'precursory'. However, at the time and in isolation none of the parameters including thermal anomalies, flank motion, anomalous degassing, seismicity,

and infrasound, were sufficiently conclusive to shed light on the events that were about to unfold.”

Reply: accepted and changes made.

3. It would only be fair to cite :

Rockfall-Avalanche and Rockslide-Avalanche Deposits at Sawtooth Ridge, Montana

M. R MUDGE

GSA Bulletin (1965) 76 (9): 1003-1014.

where definitions are given in relation to event described in the manuscript. Note the hyphen (-) in the term Rockslide-Avalanche compared to Rockslide Avalanche used in the ms. I suggest you stick with the original term and spelling coined by Mudge.

Reply: We agree that a consistent, original and proper use of terminology is required. A rockslide is defined as a fast moving landslide, actually defined by Sharpe (1938) and not by Mudge (1965), which is why we decided to not include this reference. But we carefully checked our use of terms (see also point below).

4. The authors refer to the mass movement as a “landslide”, “rockslide” or “rockslide-avalanche”. Please do not use these terms loosely as they entail different source processes and materials undergoing failure. Be sure you use the correct term and stick to it throughout the ms incl supp mats.

Reply: Accepted and changes made. As we did not further elaborate the moving products and transport physics, in the revised version we now consistently use the overarching term “landslide”. Concerning the citation, we refer to more recent review papers that have reference to this and other relevant landslide works (e.g. Siebert (1984), UI (1983), and of course Voight et al. (1985))

L65: delete “potential”

Reply: Accepted and changes made. We deleted the word as suggested

L341: delete “behaviour”; in fact the entire sentence is illogical. Is it not because these events are rare that multiparameter insights such as those reported here yield vital information on precursory processes?

Reply: Accepted and changes made. We deleted the word as suggested, and rephrased the sentence.

Congratulations on a fine paper. J Gottsmann

Reply: 😊